# Tannins amount determines whether tannase-containing bacteria are probiotic or pathogenic in IBD

Qiuyue He[1], Kenan Guo[1,2], Lulu Wang[1], Fei Xie[1], Qingyuan Zhao[1], Xianhong Jiang[1], Zhongming He[1], Peng Wang[1], Shiqiang Li[1], Yan Huang[3], Cong Zhang[1], Rongjuan Huang[1], Yang Liu[1], Fengchao Wang[2], Xiaoyang Zhou[4], Rong Niu[1], Tao Zuo[5], Yong Wang[1,*] , Chuangen Li[1,*]

**The role of dietary tannin in inflammatory bowel disease (IBD) is still not clear. Therefore, we aim to study the effect of TA in the progression of IBD. Dextran sulphate sodium (DSS)–induced model was used to mimic IBD. Metagenomics and metabolomics were performed to study the alteration of intestinal microbiota and metabolites. NCM460 and THP-1 cells were used for in vitro study. The amount of TA was associated with the outcomes of DSS-induced IBD as evidenced by in vivo and in vitro studies. Metabolomic and metagenomic analyses revealed that TA-induced enrichment of microbial metabolite gallic acid (GA) was responsible for the action of TA. Mechanistically, protective dose of GA promoted colonic mucus secretion to suppress bacterial infection and that it ameliorated DSS-induced epithelial damage by inhibiting p53 signaling, whereas toxic dose of GA directly caused epithelial damage by promoting cell cycle arrest. Therapeutic experiment showed protective dose of GA-promoted recovery of DSS-induced colonic inflammation. The role of tannase-containing bacteria can be transformed under different conditions in IBD progression.**

## Introduction

Inflammatory bowel disease (IBD) affects millions of people worldwide, including ulcerative colitis (UC) and Crohn's disease (Plevris & Lees, 2022). The main cause of IBD is the disorder of intestinal inflammatory cytokines, among which IL-1$\beta$ and IL-18 play central roles (Bauer et al, 2010). The mammalian colon coevolves with a diverse microbial ecosystem, where both the host and microbiota developed with their symbiotic interactions, partially, through shaping a robust immune system in the host (Brodin, 2022, Maynard et al, 2012). Normally, a peaceful coexistence can be kept because of the complex communication between the host and microbiota (Maynard et al, 2012). Diet is an important factor that shapes the interaction between the host and microbiota (Wastyk et al, 2021). Dietary sugar promoted DSS-induced colitis through enriching mucus-degrading bacteria *Akkermansia muciniphila* and *Bacteroides fragilis* (Khan et al, 2020). High-fat diet is a risk factor of pre-IBD (Lee et al, 2020). High-fat diet–derived free fatty acids impair the intestinal immune system and increase sensitivity to intestinal epithelial damage in a microbiota-independent manner (Tanaka et al, 2020). These previous findings imply that both intestinal microbiota and dietary metabolites have effects on IBD progression. Thus, understanding the microbial metabolism of dietary components is critical to comprehend gut ecological dynamics in response to the diet.

Tannins (TAs) are a mixture of water-soluble polyphenols with high molecular weights, contained in various kinds of plants, such as green tea, grape, and persimmon (Tomasi et al, 2022). Red wine also contains large amounts of TAs. In response to dietary compounds, a vast array of microbial enzymes was activated to produce transformed metabolites within the gut (Janiak, 2016). Among those activated enzymes, tannase was identified to be the core protein in microbiota because of their role of degrading high molecular weight tannins into subsequent downstream metabolites (Mancheno et al, 2022). Gallic acid (GA) was verified to be the main product of tannase activity over tannins compounds in most cases (Mancheno et al, 2022). In addition, TA is toxic to intestinal bacteria, and the degradation of TA into GA by activated tannase is considered to be a way of "self-protection" because the toxicity of GA is weaker than that of TA (Mancheno et al, 2022). However, the role of TA metabolism in IBD is still largely unknown.

In the present study, we investigated the effect of TA in the progression of DSS-induced IBD. We showed beneficial and pathogenic roles of tannase-containing bacteria can be transformed under different conditions, providing a new cognition for the function of

---

[1]Department of Laboratory Animal Science, Army Medical University, Chongqing, China   [2]State Key Laboratory of Trauma, Burns and Combined Injury, Institute of Combined Injury, Chongqing Engineering Research Center for Nanomedicine, College of Preventive Medicine, Army Medical University, Chongqing, China   [3]Department of Pharmaceutics, College of Pharmacy, Army Medical University, Chongqing, China   [4]Department of Biological Safety, Army Medical University, Chongqing, China   [5]Guangdong Institute of Gastroenterology, The Sixth Affiliated Hospital, Sun Yat-Sen University, Guangzhou, China

Correspondence: yongw7528@hotmail.com; lcg20050502@aliyun.com
*Yong Wang and Chuangen Li contributed equally to this work

intestinal microbiota. Our study also highlights the importance of interaction between diet and microbial enzyme in the progression of IBD, further confirming the importance of microbial metabolite in affecting intestinal disease.

# Results

## Dual effects of TA in DSS-induced IBD are associated with the amount of intake

To understand the effect of TA on IBD, mice were pretreated with different doses of TA (0, 10, 50, and 250 mg/kg for 7 d, for simplicity, the dose is thereafter referred to as TA0, TA10, TA50, and TA250). Mice were then treated with 2.5% DSS plus TA for another 3 or 7 d (Fig 1A). Among all these doses, TA50-pretreated mice showed markedly reduced body weight loss, increased colon length, and ameliorated histological change compared with TA0-treated mice at day 7 after DSS intake (Fig 1B–D). In agreement with these changes, RT–PCR showed that the inflammatory cytokines and chemokines (IL1$\beta$, IL6, TNF-$\alpha$, Cxcl1, and Cxcl2) were significantly reduced in TA50-treated mice compared with TA0-treated mice at this time point (Fig 1E). Consistently, expressions of inflammation-associated proteins, p-p65, p-ERK, and p-stat3, were remarkably reduced in TA50-treated mice (Fig 1F).

On the other hand, no difference was noted between TA0- and TA250-treated mice at day 7 after DSS intake in body weight change, colon length, and histological score. However, TA250-treated mice displayed enhanced histological change featured with an abnormal crypt structure at day 3 after DSS intake, although body weight change and colon length were comparable to TA0-treated mice (Fig 1B–D), indicating TA had dual effects on DSS-treated mice, which was associated with the amount of intake. RT–PCR and WB showed that expression of inflammatory cytokines, chemokines, and inflammation-associated proteins were comparable between TA0- and TA250-treated mice at day 3 after DSS administration (Fig S1A and B). In addition, mice consumed comparable water during DSS treatment (Fig S1C), implying that colitis sensitivity of TA-treated mice was not due to the amount of DSS intake.

Next, we examined whether pretreatment is a prerequisite for TA to exert its function. To this end, mice were administrated with 2.5% DSS plus TA for 3 or 7 d (Fig S1D). No significant difference was noted between TA-treated (both TA50 and TA250) and untreated (TA0) mice as measured by body weight change and colon length, although a slight amelioration of histological score was seen in TA50-treated mice at day 7 after DSS intake (Fig S1E–G), suggesting pretreatment of TA is a prerequisite for the effect of TA. Taken together, these observations suggest that TA has either protective or toxic effect on DSS-induced IBD and pretreatment of TA is pivotal to exert its function.

## TA intake affects intestinal barrier and displays different outcomes under germ-free and bacterial conditions

Because loss of epithelial crypts and abnormal epithelial structure were observed in TA250-treated mice at day 3 post DSS treatment

(Fig 1D), next, we examined the gamma H2A histone family member X ($\gamma$-H2AX), DNA damage marker, and tight junction proteins (TJPs), E-cadherin and occludin, markers of gut integrity. Immunohistochemical (IHC) staining displayed a reduced expression of $\gamma$-H2AX and increased expression of E-cadherin and occludin in TA50-treated mice at day 7 after DSS administration (Fig S2A). In contrast, TA250 treatment led to increased $\gamma$-H2AX expression and decreased E-cadherin and occludin at day 3 after DSS intake (Fig S2A), implying TA250 enhanced DSS-induced intestinal epithelial injury. EM analysis further confirmed altered epithelial tight junctions, including widening of spaces in the apical junctional complex and paracellular gap in TA250-treated mice (Fig S2B), suggesting TA intake led to an impaired intestinal barrier.

As TA pretreatment and non-pretreatment resulted in different outcomes, we asked whether intestinal microbiota play a role. To address this, germ-free (GF) mice were pretreated with TA for 7 d and then administrated with 2.5% DSS plus TA for another 3 or 5 d (because GF mice are more susceptible to death under DSS treatment [Hernandez-Chirlaque et al, 2016], we shortened the treatment to 5 d) (Fig S2C). Surprisingly, histological analysis showed no significant difference between TA-treated (TA50 and TA250) GF mice and control (TA0) GF mice (Fig S2D). Consistently, Western Blot assay revealed a comparable expression of $\gamma$-H2AX, TJPs, and inflammation-associated proteins (Fig S2E). These observations indicate that TA had distinct effects under GF condition compared with the observations under bacterial condition, implying intestinal microbiota are potentially involved in the action of TA.

## TA-induced bacterial alteration has no effect on DSS-treated mice

To address the role of microbiota in the action of TA, shotgun metagenomic sequencing on fecal samples collected from mice with or without TA treatment was performed (Fig S3A). Unexpectedly, no microbiota compositional discriminations were noted between TA-treated and untreated mice as ascertained by unsupervised principal components analysis (PCA) (Fig S3B). Likewise, analysis of $\alpha$-diversity using Chao-1 and Simpson indices showed no significantly altered species richness between TA-treated and untreated mice (Fig S3C). We further performed linear discriminant analysis (LDA) using the LDA effect size (LEfSe) algorithm to identify operational microbial taxa that were differentially abundant with TA intake. TA50 administration led to the reduction of two main genera, *Prevotella* and *Bacteroides*, whereas TA250 administration mainly reduced the genus of *Lactobacillus* (Fig S3D). RT–PCR further confirmed the change of those bacteria (Fig S3E). Next, to assess the role of these altered bacteria in the effect of TA, GF mice were orally administrated with feces from mice subjected to TA treatments described above (for simplicity, referred to as GF-TA0, GF-TA50, and GF-TA250, respectively) for 14 d followed by administration of DSS for 3 or 7 d (Fig S4A). RT–PCR was performed to validate bacterial colonization (Fig S4B). No differences were noted in body weight change, colon length, and histological change between these mice with fecal microbiota transplantation (FMT) (Fig S4C–E). Taken together, these results demonstrate that TA could not change the composition of intestinal microbiota, and TA-induced alteration of some species of bacteria has no effect on DSS-induced IBD.

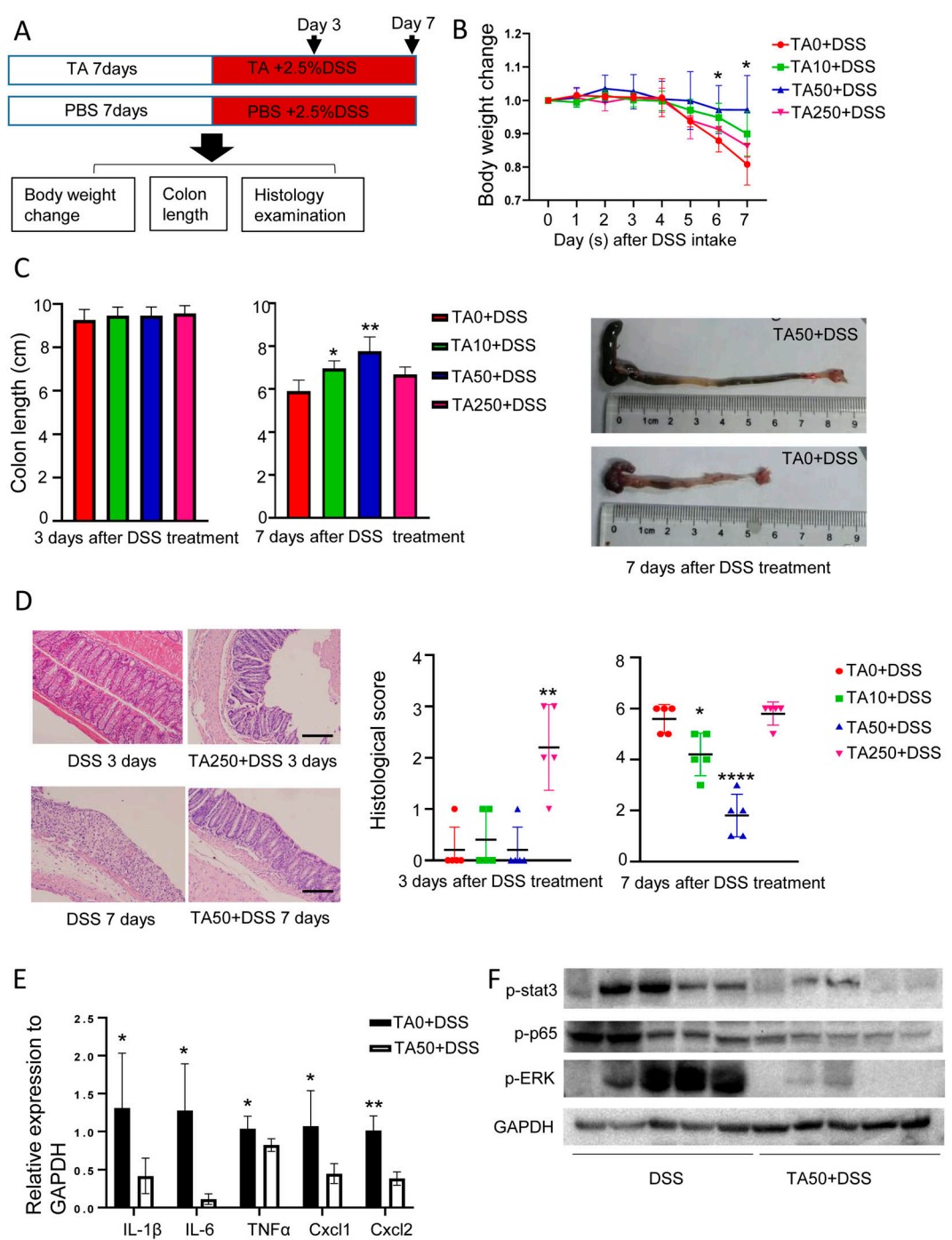

**Figure 1.  TA displays dual effects in DSS-induced IBD.**
Mice were treated with different doses of TA for 7 d and then 2.5% DSS plus TA were co-treated to mice for another 3 or 7 d (n = 5/group). **(A)** Method for mice treatment. **(B)** Body weight change of TA-pretreated mice in the presence of DSS. **(C)** Colon length of mice pretreated with TA. **(D)** Histological change of mice pretreated with TA. Scale bar, 100 $\mu$m. **(E)** Change of inflammatory cytokines and chemokines of colon tissues with TA50 pretreatment after 7-d DSS intake. **(F)** Expression of inflammation-associated proteins of colon tissues with TA50 pretreatment at day 3 post DSS administration. *P < 0.05, **P < 0.01, ****P < 0.0001.
Source data are available for this figure.

## TA-induced enrichment of microbial metabolite gallic acid mimics the effect of TA in vitro

Because the effect of TA on DSS-induced IBD was not due to the change of intestinal microbiota, therefore, we asked whether TA induces alteration of metabolites in feces. To this end, fecal samples, collected from the same mice used for metagenomic analysis, were performed metabolomic analysis. Although no difference was noted in the composition of intestinal metabolites as determined by PCA (Fig 2A), volcano plot assay revealed that several

metabolites were enriched with TA administration (Fig 2B and Table S1). In TA250- and TA50-treated mice, 41 metabolites and 12 metabolites were enriched, respectively (Table S1). Because TA250 treatment induced colonic injury in mice, we examined the effect of mostly enriched metabolites (FC > 3.5, VIP > 1, $P$ < 0.05) on NCM460 cells, including stearoyl ethanolamide, LPE17:0, GA, and 10-nitro-linoleate. Surprisingly, only GA, which was enriched in feces of both TA50- and TA250-treated mice, displayed dual effects. MTT assay revealed that GA enhanced DSS (2.5%)-induced cellular injury at the dose of 100 and 50 μg/ml compared with DSS-alone–treated cells, whereas GA (20 and 10 μg/ml) protected cells from DSS-induced injury (Fig 2C), resembling the observations in TA-treated mice. This phenomenon was not seen in other tested metabolites. Furthermore, immunofluorescent assay also proved that GA (50 μg/ml) notably reduced the expression of E-cadherin. In contrast, GA (10 μg/ml) protected the intact of E-cadherin (Fig 2D). However, TA did not show significant effect on DSS-induced injury in NCM460 cells, although a slight cellular damage was seen at day 2 with the dose of 100 and 50 μg/ml (Fig S5A and B). These observations indicate that GA is responsible for the effect of TA, and meanwhile GA harbors stronger biological activity than TA.

Evidence has shown that tannase is a core enzyme of intestinal microbiota (Mancheno et al, 2022). TA treatment always activates bacterial tannase to produce GA as the main downstream metabolite (Mancheno et al, 2022; Ristinmaa et al, 2022). So, next, we examined the activity of tannase in TA-treated feces. Tannase activity was significantly increased with TA50 and TA250 administration (Fig 2E). HPLC further confirmed the enrichment of GA in feces and colon tissues of TA-treated mice (Fig 2F). To further investigate the relationship between TA and GA, feces collected from WT mice (SPF environment) were cultured in vitro using nutrient broth medium in the presence of TA (0, 100, and 50 μg/ml) (Fig S5C). At day 1 and 5, supernatant was collected for quantifying GA. It showed that GA was markedly enriched in the presence of TA, which was dose- and time-dependent (Fig S5D). Because FMT results showed no difference between GF-TA0 and GF-TA50/250 under DSS administration, we examined GA in faces and tissue samples collected from mice for FMT via HPLC. GA was failed to be detected in these samples. Taken together, these results suggest that TA exerts its action through producing GA.

### GA recapitulates the effect of TA in vivo

The above observations led us to examine the effect of GA in vivo. Because of the high transformation efficiency of TA, the same doses of GA (0, 10, 50, and 250 mg/kg, for simplicity, referred to as GA0, GA10, GA50, and GA250, respectively) were administered to mice 7 d before or at the same time with DSS administration (Fig 3A). In line with the observations of TA-treated mice, pretreatment of GA also displayed dual effects depending on the amount of intake. Mice treated with GA50 showed mostly ameliorated body weight loss, histological change, and significantly increased colon length after 7-d DSS treatment (Fig 3B–D). GA250 treatment showed a toxic effect on mice featured with an abnormal crypt structure at day 3 after DSS administration, although the body weight and colon length were comparable (Fig 3B–D). These differences were not observed in mice without pretreatment of GA, although a slight amelioration of histological score was noted (Fig S6A–C). RT–PCR showed that

GA50 markedly inhibited the expression of inflammatory cytokines and chemokines at day 7 post DSS administration (Fig 3E). Consistently, WB assay further revealed that GA50 treatment down-regulated p-ERK, p-p65, and p-Stat3 compared with GA0 treatment at day 7 after DSS intake (Fig 3F). Although enhanced histological change was noted at day 3 after DSS administration in GA250-treated mice, the level of inflammatory cytokines and chemokines were not remarkably changed as evidenced by RT–PCR and WB at this time point (Fig S6D and E). Of note, GA250-treated mice displayed a markedly reduced colon length at day 7 after DSS intake in both GA-pretreated and non-pretreated mice (Figs 3D and S6B). In line with this, RT–PCR and WB revealed upregulation of inflammatory cytokines, chemokines, and inflammation-associated proteins in GA250-treated mice at this time point (Figs 3G and S6F). Because bacterial penetration was pivotal to overt intestinal inflammation, we examined the presence of bacteria in colon tissues by staining with fluorescent in situ hybridization of universal bacterial 16S rRNA. At day 3 post DSS treatment, no bacteria invasion was detected in neither GA250- nor GA0-treated mice (Fig S6G), which was consistent with the parallel intestinal inflammation in these mice with 3-d DSS intake. At day 7 post DSS treatment, similar bacteria invasion was observed in both GA250- and GA0-treated mice (Fig S6G), indicating other factors were involved in the enhanced inflammation in GA250-treated mice with 7-d DSS administration.

We also analyzed the composition of intestinal microbiota and metabolites using feces with or without GA intake through 16S ribosomal RNA (rRNA) gene sequencing analyses and metabolomics. For bacteria, α-diversity showed no significant change of species richness as determined by Shannon and Simpson indices between GA-treated and untreated mice (Fig S7A). Likewise, β-diversity of microbial population in GA-treated groups was not altered as reflected by a distinct clustering pattern on unweighted principal coordinates analysis (PCoA) (Fig S7B). For the metabolites, no difference was noted between GA-treated and untreated mice as determined by PCA assay (Fig S7C). These observations further implicated that GA resembled the effect of TA. Furthermore, we also noticed that metabolites displayed an opposite change between GA50- and GA250-treated mice, which may partially explain their opposite effects (Fig S7D and Table S2) Taken together, these observations indicate that GA generally recapitulates the effect of TA in vivo, although some differences were also seen.

### GA exhibits both beneficial and detrimental effects in GF mice depending on the amount of intake

Because pretreatment of GA displayed stronger effect than non-pretreated mice, next, we performed the cell experiment to see whether this observation can be recapitulated in vitro. To address this, NCM460 cells were pretreated with GA (0, 10, and 50 μg/ml) for overnight, and then 2.5% DSS was added to cells for another 24 h. MTT assay showed no difference between GA-pretreated and non-pretreated cells (Fig S7E), implying other factors contribute to the action of GA in vivo.

As GA has stronger biological activity than TA, which showed limited action under GF condition, to verify the effect of GA in the GF environment, GF mice were pretreated with GA50 and GA250 for 7 d and then treated with 2.5% DSS plus GA for 3 or 5 d (Fig 4A). We

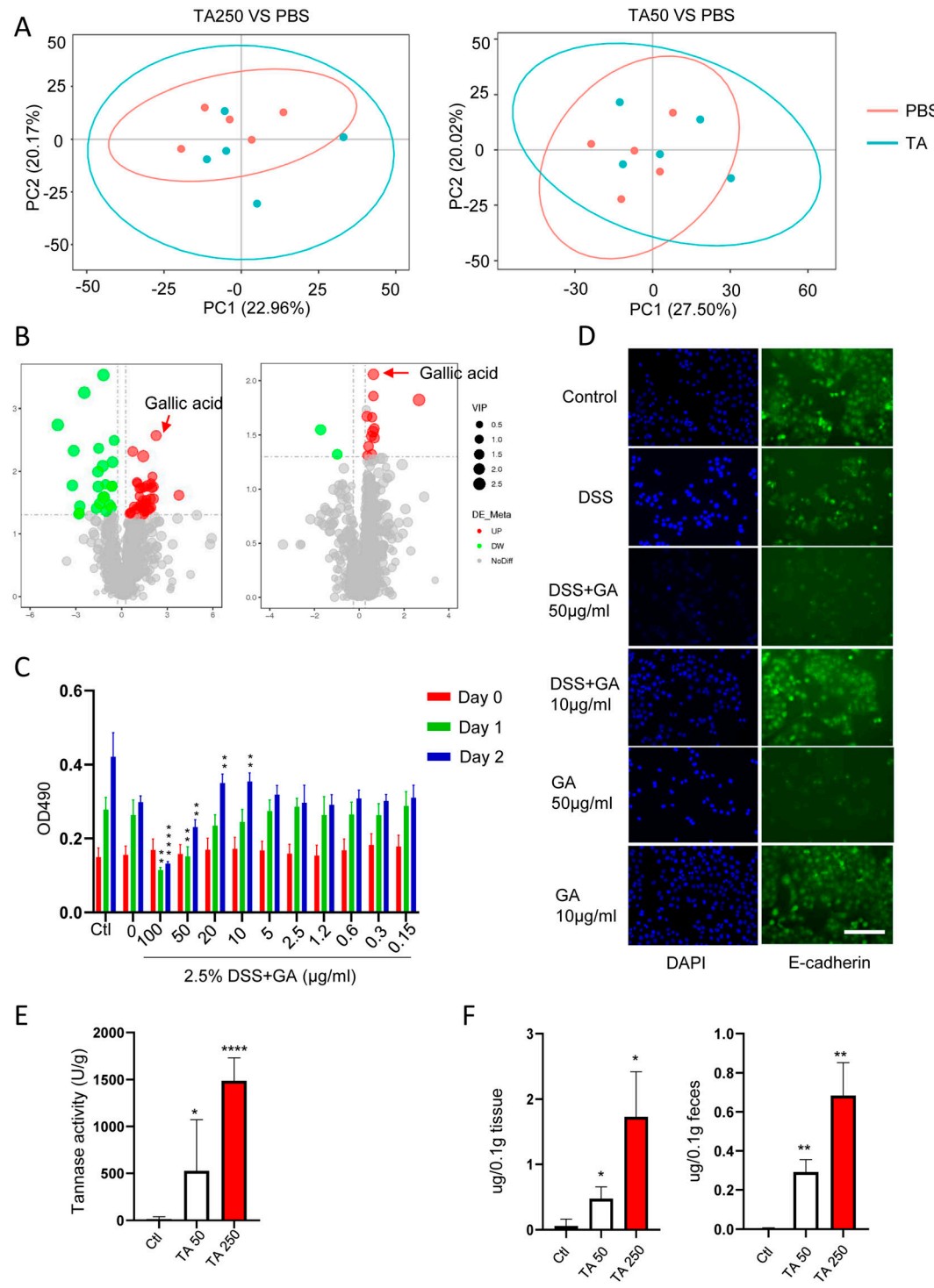

**Figure 2. TA-induced gallic acid enrichment is responsible for the action of TA.**
Feces with TA treatment were collected for metabolomics (n = 5/group). **(A)** Principal component analysis of metabolite composition between TA-treated and untreated mice. **(B)** Volcano plot assay for notably altered metabolites. **(C)** MTT assay for NCM460 cells treated with GA and DSS. **(D)** IFA staining of E-cadherin for NCM460 cells with GA and DSS treatment. Scale bar, 50 μm. **(E)** Measurement of tannase activity using feces of mice with TA treatment. **(F)** HPLC assay for GA volume measurement. *P < 0.05, **P < 0.01, ****P < 0.0001. Source data are available for this figure.

observed that GA0- and GA250-treated GF mice became weak featured with reduced motility at day 4 and died at day 5 after DSS intake, whereas GA50 administration prolonged the survival of

treated mice to day 7 (Fig 4B). HE staining revealed that GA50-treated GF mice displayed ameliorated colonic damage compared with control GF mice after 5-d DSS treatment, whereas GA250

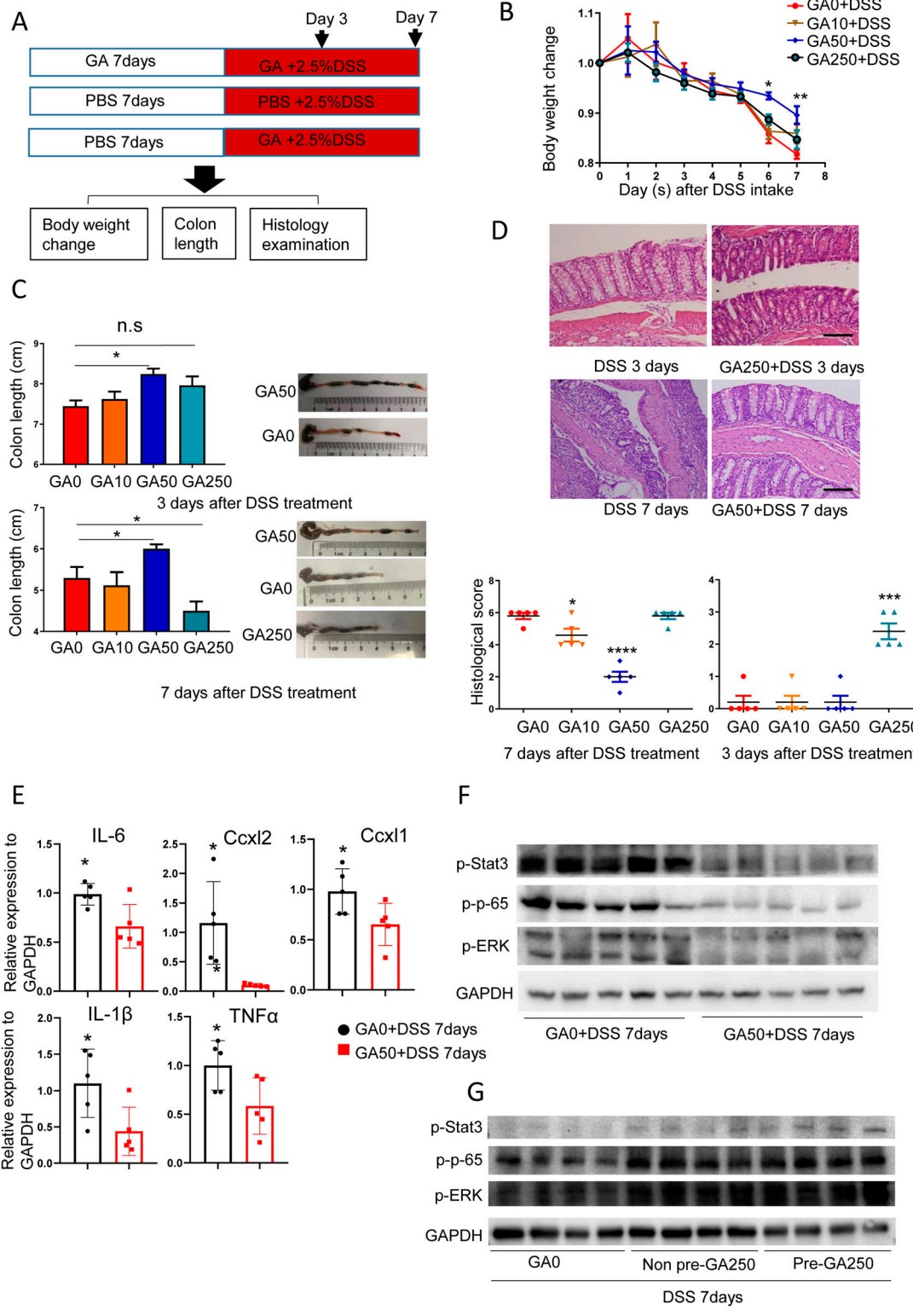

**Figure 3. GA resembles the observations of TA-treated mice.**
Mice were treated with different doses of GA before or together with DSS intake (n = 5/group). **(A)** Method for mice treatment. **(B)** Body weight change of mice treated with different doses of GA. **(C)** Colon length of mice treated with different doses of GA. **(D)** Histological change of mice. Scale bar, 100 μm. **(E)** Inflammatory cytokines and chemokines change of GA0- and GA50-treated mice at day 7 after DSS intake. **(F)** Examination of inflammation-associated proteins of GA0- and GA50-treated mice at day 7 after DSS intake by WB. **(G)** Examination of inflammation-associated proteins of GA0- and GA250-treated mice at day 7 after DSS intake by WB. *P < 0.05, **P < 0.01, ****P < 0.0001. Source data are available for this figure.

administration enhanced DSS-induced colonic damage (Fig 4C). IHC staining further confirmed that GA50 treatment reduced γ-H2AX expression in concomitant with the increased expression of E-cadherin, which showed an opposite trend in GF mice with GA250 administration (Fig 4D). These results indicate that GA displays dual effects on DSS-induced colonic damage, which is associated with the amount of intake. Moreover, these observations further prove GA possesses stronger biological activity than TA.

### GA affects colonic mucus secretion through regulating goblet cells

Previous studies proved that GF mice are more susceptible to DSS administration due to the insufficient inner mucus secretion (Johansson et al, 2014). Our results revealed a notably delayed death time of GA50-treated GF mice exposed to DSS. Therefore, we asked whether GA could affect mucus secretion. To address this, mice were treated with GA50 or PBS for 7 d, and then mice were treated with GA50 plus DSS for different time points (0, 3, and 5 d) (Fig 5A). The colon tissues were fixed in Carnoy's fixative solution to preserve the mucus layer, and mucin layer thickness was measured by immunostaining of Muc2. PAS–alcian blue staining was performed to detect pre-goblet and goblet cells in the colon. In the absence of DSS, GA50 treatment increased the staining intensity of goblet cell granules and the thickness of mucin layer compared with control mice, suggesting that GA50 treatment could promote differentiation of goblet cells and mucin secretion (Fig 5B). In the presence of DSS, GA50 treatment markedly reduced the loss of mucus layer compared with GA0-treated mice at day 0 and 3 after DSS intake (Fig 5C). At day 5 post DSS administration, bacterial translocation into the colonic mucosa was not detected in GA50-treated mice, whereas an apparent bacterial infiltration into the mucosal tissue of GA0-treated mice was seen, although mucus was barely detected in both groups (Fig 5D). Moreover, the goblet cells in the colon of GA250-treated mice were markedly reduced at day 3 after DSS intake as determined by PAS–alcian blue staining (Fig 5D).

Inner mucus is critical in preventing invasion of bacteria into intestinal epithelium (Johansson et al, 2014). Bacterial invasion always induces overt inflammatory response, which is considered as the initial step of several intestinal diseases, such as IBD and CRC (Hernandez-Chirlaque et al, 2016; Coleman et al, 2018). The above results showed that GA250 treatment damaged goblet cells and reduced mucus secretion. This led us to test whether prolonged GA250 treatment can induce colonic inflammation. To address this, mice were treated with GA250 for 30 d (Fig 5E). RT–PCR revealed that inflammatory cytokines were notably induced in GA250-treated mice compared with control mice (Fig 5F). WB assay demonstrated that p-stat3, p-p65, and p-ERK were upregulated with GA250 treatment (Fig 5G). Furthermore, reduced thickness of Muc2 in concomitant with bacterial penetration was observed in mice with GA250 treatment (Fig 5G). These results indicate that GA could regulate mucus secretion to inhibit or induce colonic inflammation.

### GA shows distinct effects in epithelial cells and macrophages

Next, we examined whether GA had a direct effect on inflammation. To this end, NCM460 cells (epithelial cell line) and THP-1 cells (macrophage cell line) were treated with DSS and LPS, respectively, with or without GA administration. GA (both 50 and 10 μg/ml) exhibited anti-inflammatory effect in DSS-treated NCM460 cells (Fig S8A and B). In contrast, the toxic dose of GA (50 μg/ml) showed synergistic effect with LPS to induce inflammatory cytokines, whereas the beneficial dose of GA (10 μg/ml) had no effect on LPS-induced inflammation in THP-1 cells (Fig S8C and D), indicating the effect of GA on epithelial cells was distinct from those on macrophages. To explore whether GA could induce inflammation directly in THP-1 cells at the dose of 50 μg/ml, we treated THP-1 cells with GA for overnight. RT–PCR showed no significant difference in expression of inflammatory cytokines between GA-treated and untreated cells (Fig S8E). Consistently, GF mice treated with GA (GA0, GA50, and GA250) also displayed comparable levels of inflammatory cytokines, chemokines, and inflammation-associated proteins as evidenced by RT–PCR and WB assays (Fig S8F), suggesting the presence of bacteria is a prerequisite for GA to regulate colonic inflammation. Collectively, these observations imply that GA plays different roles in epithelial cells and macrophages.

### Protective dose of GA ameliorates DSS-induced cellular damage through inhibiting p53 signaling, whereas toxic dose of GA induces cell death by arresting cell cycle

Because we observed that GA had a direct effect on DSS-induced epithelial injury, we further explored the potential mechanism for GA to exert its function. To this end, RNA sequencing was performed using NCM460 cells in the presence of DSS with or without GA (10 μg/ml) treatment. A number of differentially expressed genes were identified in cells. To better understand the targeted pathways, Kyoto Encyclopedia of Genes and Genomes (KEGG) pathway analysis was performed. DSS administration significantly inhibited the ribosome pathway, whereas activated virus-associated pathways (influenza A, hepatitis C, measles, Kaposi sarcoma-associated herpesvirus infection, herpes simplex infection, human papillomavirus infection, and hepatitis B) and inflammation-associated pathways (JAK-STAT signaling pathway, Toll-like receptor signaling pathway, TNF signaling pathway, and NOD-like receptor signaling pathway) (Fig 6A and Table S3). In contrast, in the presence of GA (10 μg/ml), pathogen and inflammation-associated pathways (influenza A, herpes simplex infection, Epstein–Barr virus infection, measles, hepatitis C, and NOD-like receptor signaling pathway) were notably downregulated (Fig 6B and Table S4). Moreover, we also noticed that GA (10 μg/ml) administration led to marked downregulation of the p53 signaling pathway, which is tightly associated with cellular injury. Because dysregulation of ribosome and upregulation of inflammation could result in the activation of p53 signaling (Lindstrom et al, 2022), we examined the expression of p53 and p-p53 in cells with DSS treatment. WB assay showed that p-p53 was upregulated in DSS-treated cells, whereas p53 was comparable between DSS-treated and untreated cells (Fig 6C). MTT assay also revealed that p53 inhibitor, pifithrin-α (PFTα), significantly increased cell viability (Fig 6D), indicating p53 plays a pivotal role in DSS-induced cellular injury. Downstream proteins of the p53 pathway, BCL2, and Bax were also suppressed by GA (Fig 6C). In the presence of DSS, toxic dose of GA (50 μg/ml) displayed downregulated pathogen and inflammation-associated pathways, which

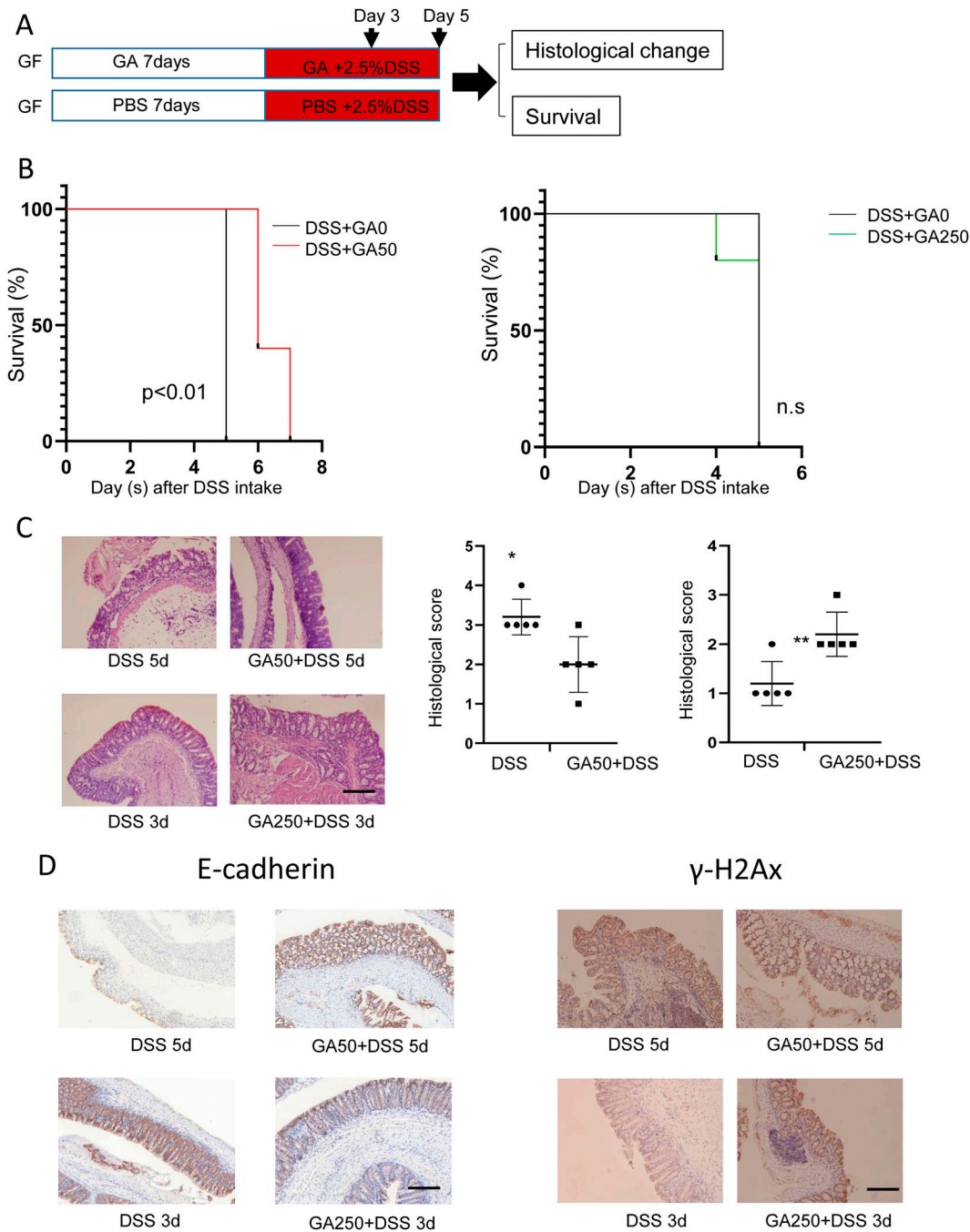

**Figure 4. GA displays beneficial and toxic action in GF mice.**
GF mice were treated with GA50 and GA250 in the presence of DSS (n = 5/group). **(A)** Method for mice treatment. **(B)** Survival of GF mice with GA50 and GA250 treatment. **(C)** Histological change of GF mice. Scale bar, 100 µm. **(D)** Expression of E-cadherin and Y-H2AX in colon tissues with or without GA administration.
Source data are available for this figure.

is consistent with the RT–PCR result of NCM460 cells treated with DSS plus toxic dose of GA, whereas the ribosome and breast cancer pathways were upregulated (Fig S9A and Table S5)

As toxic dose of GA alone could induce severe cellular damage, to explore the potential mechanism, RNA sequencing was performed using NCM460 cells with or without GA (50 µg/ml) treatment. KEGG

pathway analysis revealed that GA (50 µg/ml) mainly promoted cell cycle arrest to cause cellular damage (Fig 6E and Table S6). Cell cycle assay showed that administration of GA (50 µg/ml) increased S-phase cell population in concomitant with decreased G0/G1 phase cells (Fig 6F). RT–PCR further confirmed that expression of cell cycle-associated genes were notably downregulated with GA

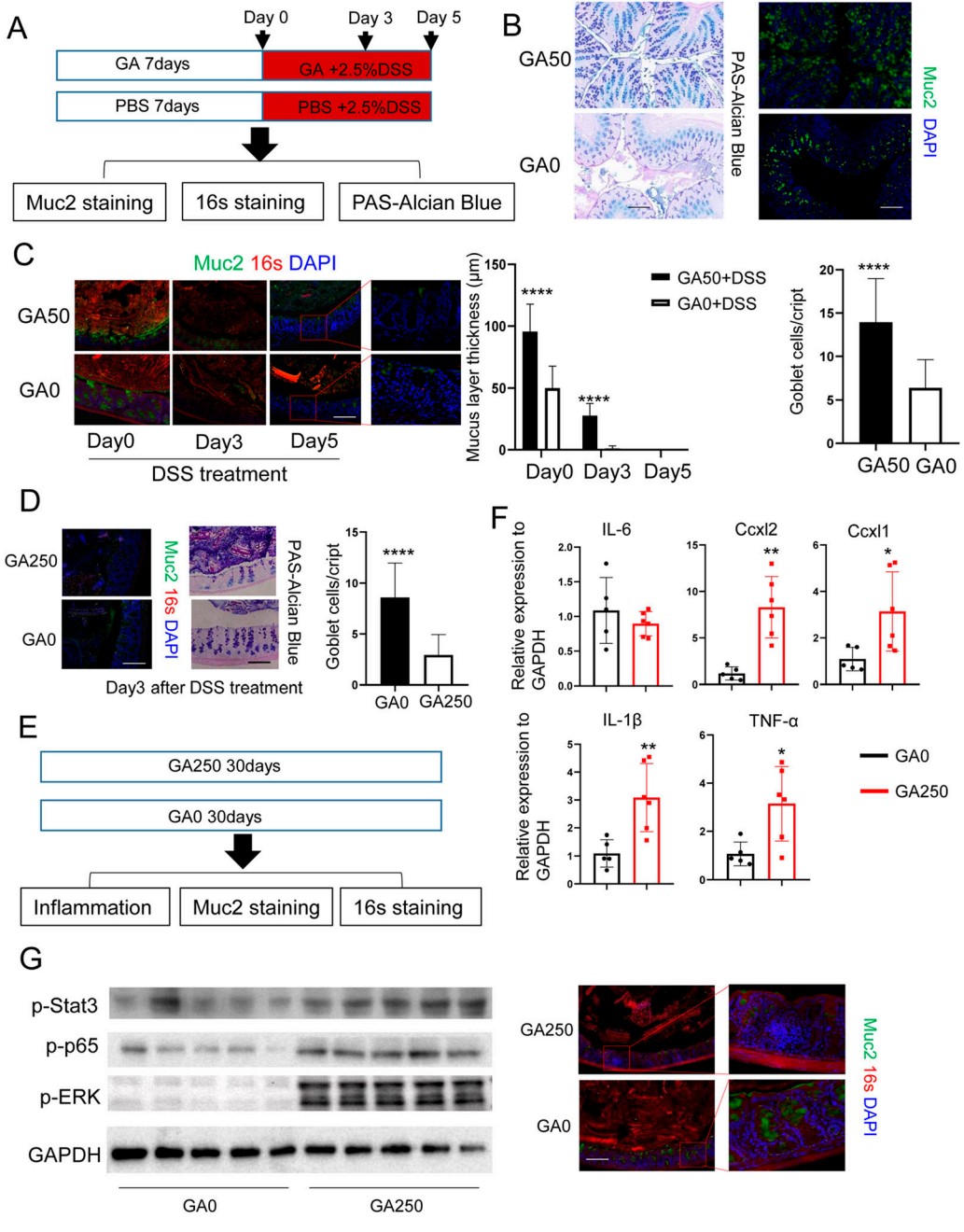

**Figure 5. GA exerts its function mainly through affecting goblet cells.**
Mice were fed with GA50 and GA250 and euthanized at different time points (n = 5/group). **(A)** Method for mice treatment. **(B)** Muc2 mucin of WT mice was stained with PAS–alcine blue and IFA staining. Scale bar, 100 μm. **(C)** Mucin layer and intestinal bacteria of GA50-treated mice were stained using anti-Muc2 antibody and fluorescent in situ hybridization probe for universal bacterial 16S rRNA. Scale bar, 50 μm. **(D)** Muc2 mucin of GA250-treated mice was stained with PAS–alcine blue and IFA staining. **(E)** Method for long-term GA250 treatment of mice. **(F)** Inflammatory cytokines and chemokines examination by RT–PCR. **(G)** Inflammation-associated proteins examination and mucin layer and intestinal bacteria assessment of long-term GA250-treated mice. Scale bar, 50 μm. *P < 0.05, **P < 0.01, ****P < 0.0001. Source data are available for this figure.

administration (Fig 6G). Moreover, we noticed that GA (50 μg/ml) treatment promoted cytokine–cytokine receptor interaction (Fig 6G). Because we observed GA250 enhanced inflammatory cytokines production at day 7 after DSS intake, which was not associated with bacterial invasion, we examined the effect of GA (50 μg/ml) on the expression of cytokine receptor in macrophages by RT–PCR.

It showed significantly increased levels of IL-6R, IL-1R1, CxCR2, and TNFRSF1B in THP-1 cells with GA treatment (Fig S9B). Taken together, these observations indicate that beneficial dose of GA inhibits p53 signaling to protect DSS-induced cellular injury, whereas toxic dose of GA promoted cell cycle arrest to induce cellular damage.

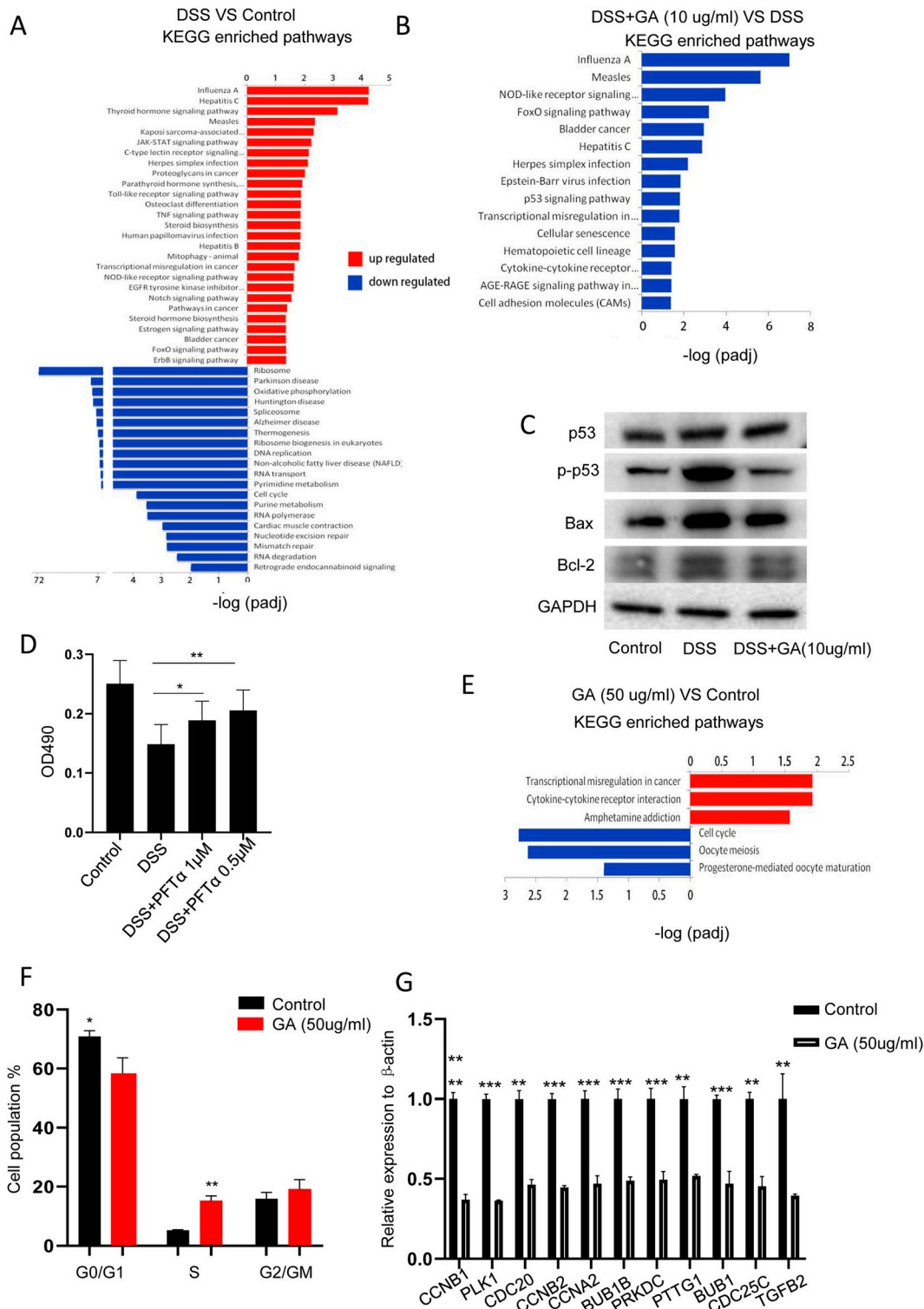

**Figure 6. Beneficial dose of GA protected DSS-induced cellular damage through suppressing p53 signaling, whereas toxic dose of GA caused cell damage by affecting cell cycle.**

NCM460 cells were treated with GA (50 and 10 μg/ml) with or without DSS. Then, RNA-Seq was performed. **(A)** KEGG enriched pathway analysis of DSS-treated and control cells. **(B)** KEGG enriched pathway analysis of DSS plus beneficial dose of GA (10 μg/ml) and DSS-treated cells. **(C)** WB examination of p53 and its downstream proteins. **(D)** MTT assay for evaluating the role of p53 in DSS-induced cellular damage. **(E)** KEGG enriched pathway analysis of toxic dose of GA (50 μg/ml) and control cells. **(F)** Cell cycle assay of toxic dose of GA (50 μg/ml) and control cells. **(G)** Cell cycle–associated genes examination by RT–PCR. *P < 0.05, **P < 0.01, ***P < 0.001. Source data are available for this figure.

## Protective dose of GA (GA50) promotes recovery of acute colonic inflammation

The above findings led us to study whether GA has therapeutic potential for intestinal inflammation recovery. To this end, mice were administrated with 2.5% DSS for 7 d, then DSS was removed and mice were treated with GA50 or GA0 orally. Mice were euthanized at day 3 and 7 after GA treatment (Fig 7A). In the control group, 8 out of 15 mice died between day 0 and 3 post GA treatment, whereas only 3 mice died in the GA50-treated group. GA50-treated mice showed significantly reduced body weight loss compared with GA0-treated mice, especially in the first 3 d after GA intake (Fig 7B). At day 3 after GA intake, 4 mice in each group were euthanized for further analysis. Histological analysis revealed that GA50 markedly reduced colonic epithelial inflammation at day 3 post GA treatment (Fig 7C). Consistent with those observations, inflammatory cytokines were notably reduced in GA50-treated mice compared with GA0-treated mice at day 3 after GA intake (Fig 7D). WB assay revealed that GA50 administration suppressed the expression of inflammation-associated proteins, among which p-p65 showed the most reduction (Fig 7E). The phenotypic differences between GA0- and GA50-treated mice disappeared at day 7 post GA intake because the colonic epithelium in both groups almost recovered to normal (Fig S10A–C). To further assess the effect of a long-term GA50 treatment, we treated mice with GA50 for a month, then inflammatory cytokines, chemokines, TJPs, cell proliferation, and mucus-associated genes were examined by RT–PCR. Only Muc2 was significantly increased with GA50 treatment (Fig S10D), indicating GA50 is a safe dose in mice. Taken together, these results indicate that protective dose of GA is capable of promoting recovery from colonic inflammation.

## Discussion

The intestine is a home of trillions of microorganisms, the number of which increases progressively along the gastrointestinal tract. Microorganisms are sparse in the proximal region and dense at the distal region of the gut (Lynch & Pedersen, 2016). Microorganisms and their metabolites have diverse functions. Commensal bacteria help educate the immune system, protect against pathogen colonization, and participate in nutrient metabolism (van de Wouw et al, 2017; Wastyk et al, 2021). Interactions between microorganisms and diet have a notable impact on the progression of IBD. Tannins are important dietary components, which possess diverse bioactivities. Tannase is a core enzyme in human microbiota, which degrades TA into downstream metabolites, among which GA is the main one in most cases (Mancheno et al, 2022). Here, we found that TA exerts its function through producing GA by intestinal microbiota in a dose- and time-dependent manner. In addition to bacterial tannase, another bacterial enzyme was found to be able to produce GA. For example, GA can be produced through the activation of bacterial enzyme shikimate dehydrogenase (SDH) in response to 3-dehydroshikimate (Muir et al, 2011). These findings highlight the importance of bacterial enzyme in dietary metabolism and the progression of IBD. Our study revealed that TA had no significant effect on the composition of intestinal microbiota. However,

Kitabatake et al previously showed that persimmon-derived tannin notably altered microbial composition (Kitabatake et al, 2021). Because they used female BALB/c mice, which is different from the mice used in our study, the gender and animal species may influence the action of TA. Moreover, derivation of TA may be another factor to affect its action.

Although DSS treatment activated p-Stat3, inflammation was barely induced because of the lack of bacteria invasion, suggesting bacteria penetration is pivotal in the induction of inflammation (Hernandez-Chirlaque et al, 2016; Coleman et al, 2018). Here, we also observed that the presence of intestinal microbiota is essential for the enhanced inflammation in mice with toxic dose of GA treatment. Moreover, goblet cells play critical roles in protecting the intestine. Once bacteria invade the colonic epithelium, goblet cells would secret MUC2 mucin to "wash away" the invaded bacteria (Birchenough et al, 2016). Here, we found that a beneficial dose of GA promoted differentiation of goblet cells to produce more inner mucus, suppressing invasion of intestinal microorganisms. Because we did not detect a direct inflammation inhibition by GA in macrophages, we assumed that GA may exert its protective function mainly through suppressing bacterial invasion by promoting mucin secretion in vivo.

Furthermore, beneficial dose of GA inhibited DSS-induced cellular injury and reduction of TJPs. It is well known that intact of intestinal epithelium is pivotal to inhibiting the penetration of bacteria (Johansson et al, 2014; Zeisel et al, 2019). Inflammatory cytokines, such as IL-1$\beta$, increased intestinal permeability by breaking the tight junction barrier (Rawat et al, 2020). So, it is reasonable that beneficial dose of GA protects the intact of epithelium through suppressing epithelial inflammation. These observations suggest that protective dose of GA strengthens the intestinal barrier by promoting mucus secretion and directly protecting epithelial cells, which facilitates recovery of colonic inflammation. Moreover, we also noticed that beneficial dose of GA suppressed multiple virus infection pathways. Because virus is another important microorganism in feces (Nakatsu et al, 2018; Lam et al, 2022), inhibition of virus infection may also contribute to the protective effect of GA. Because intestinal bacterial translocation was always seen in other types of cancer (Hueso et al, 2020; Parida et al, 2021), resulting in complications that negatively influence patients' outcomes, our study provides a way of intestinal barrier protection to improve the outcomes of patients. One limitation of our study is that we did not further explore the mechanism of GA in promoting mucus secretion.

In conclusion, our data suggest that the amount of TA intake is pivotal to determining the characteristics of tannase-containing bacteria, which notably affects the progression of digestive disease. Beneficial dose of GA protected intact of the intestine through two ways: promoting mucus secretion and directly suppressed damage of epithelial cells.

## Materials and Methods

### Cell culture

NCM460 and THP-1 cell lines were purchased from American Type Culture Collection. Cells were cultured in DMEM (11885084; Gibco) or Roswell Park Memorial Institute 1640 (22400089; Gibco)

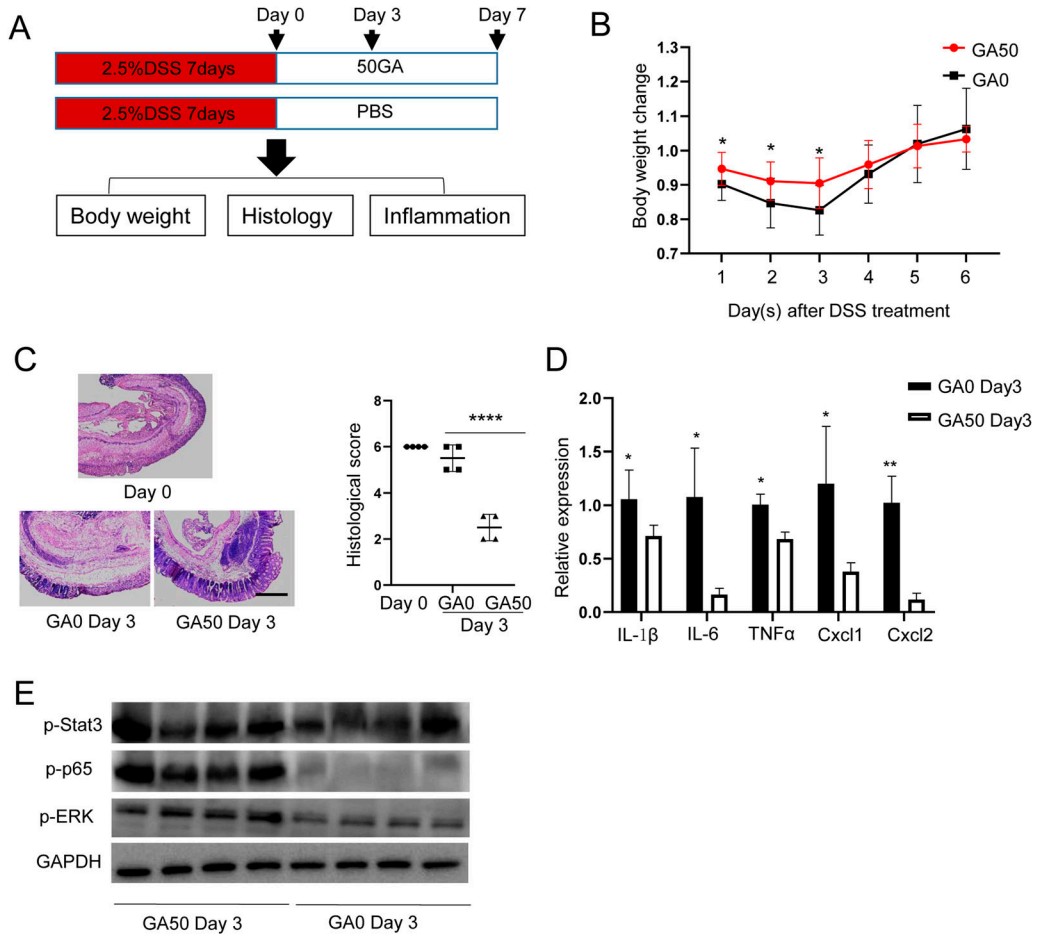

**Figure 7. GA50 promotes recovery of DSS-induced IBD.**
Mice were treated with DSS for 7 d, and then DSS was removed and GA50 was administrated to mice orally. Mice were euthanized at day 0, 3, and 7. **(A)** Method for mice treatment. **(B)** Body weight change of mice with or without GA50 treatment. **(C)** Histological analysis of mice with or without GA50 treatment. Scale bar, 100 $\mu$m. **(D)** Chemokines and inflammatory cytokines examination by RT–PCR. **(E)** Expression of inflammation-associated proteins. *$P < 0.05$, **$P < 0.01$, ****$P < 0.0001$. Source data are available for this figure.

supplemented with 10% fetal bovine serum (10099-141C; Gibco) and maintained at 37°C in a humidified incubator with 5% $CO_2$.

### Cell viability assay

Cells ($10^3$ cells/well) were seeded into 96-well plates. Different doses of GA and TA (0.1, 0.05, 0.02, and 0.01 mg/ml) were added to cells in the presence of 2.5% DSS or not. MTT (298-93-1, Merck) (5 mg/ml) was then added to cells for 4 h. After that, 100 $\mu$l of DMSO was added to cells, and absorbance was measured at 490 nm using a microplate reader (Thermo Fisher Scientific). All experiments were conducted three times in triplicate.

### Cell cycle assay

Cells ($10^6$ cells/well) were seeded into 12-well plates. Different doses of GA (0.5 and 0.02 mg/ml) was added to cells in the presence of 2.5% DSS or not. 24 h later, cells were fixed with cold 70% ethanol for overnight. Cells were then stained with FxCycle PI/RNase staining solution (F10797; Life technologies) for 30 min. Cells were finally analyzed with

flow cytometry. The cell cycle was calculated by FlowJo 7.6. All experiments were conducted three times in triplicate.

### RNA extraction reverse transcription and real-time PCR (RT–PCR)

RNA was extracted from cells or colonic tissues using GeneJET RNA Purification Kit (K0732; Thermo Fisher Scientific), according to the manufactures' instruction. RNA was quantified by NanoDrop 1000 (Thermo Fisher Scientific), and then 1 $\mu$g of RNA was applied for transcription using RevertAid Fist Strand cDNA Synthesis Kit (K1622; Thermo Fisher Scientific), according to the instruction. A $\Delta\Delta$Ct method was used for the calculation of genes' expression relative to GAPDH. The primers are summarized in Table S7-1.

### Western blot

Protein of cells or tissues was extracted with RIPA buffer (9806S; Cell Signaling Technology) supplemented with Halt Protease and Phosphatase Inhibitor Cocktail (UK292403; Thermo Fisher Scientific). Protein was quantified using BCA Protein Assay Kit

(P0012S; Beyotime). Then, 20 μg total protein was used for SDS–PAGE. Afterward, the protein was transferred to a poly-vinylidene fluoride membrane (03010040001; Roche). After transfer, the membrane was incubated with 3% BSA (A8020; Solarbio). Primary antibody was then incubated at 4°C overnight, after which the membrane was washed with 0.1% TBST for three times (10 min for each time). After incubation with secondary antibody for 1 h, the membrane was washed for three times again. Protein bands were detected using SuperSignal West Pico PLUS Chemiluminescent Substrate (W1331524; Thermo Fisher Scientific) and recorded with the ChemiDoc MP Imaging System (Bio-Rad). Antibodies are summarized in Table S7-2.

### IHC staining

After tissue sections were dewaxed, the antigen was retrieved with antigen unmasking solution. Then, the tissues were blocked with 5% goat serum (BMS0050; Abbkine), followed by incubating with primary antibodies as recommended by in-struction in a humidified chamber at 4°C for overnight. On the second day, biotinylated secondary antibody was added on the slides and incubated in a humidified chamber at room tem-perature for 30 min. After that, diluted Sav-HRP conjugates (405210; Biolegend) were applied appropriately to slides and incubated in a humidified chamber at room temperature. DAB substrate solution (926901; Biolegend) was used to incubate to the slides for 30 min and later allowed the color development for less than 5 min until the desired color intensity was reached. After washing, the slides were immersed in hematoxylin (H3136; Sigma-Aldrich) for 1–2 min. The slides were observed after dehydration and mounting.

### Immunofluorescence staining

Cells were seeded into six-well plates. After treatment, the cells were fixed with 4% PFA for 20 min. Then, the cells were per-meabilized for 5 min in PBS with 0.25% Triton X-100 (T8787; Sigma-Aldrich). After blocking with 5% goat serum for 30 min, the cells were incubated overnight at 4°C with a primary antibody in a humidified box. The tissues were treated with a secondary antibody (Alexa Fluor 488–labeled donkey anti-rabbit antibody; Invitrogen) for 1 h, after washing for three times with 0.1% TBST. Finally, DAPI (P0131; Beyotime) was added to the cells to stain the cellular nucleus.

### FISH of 16S rRNA and mucin staining

Eubacterial (universal) 16S rRNA probe tagged with Cy3 (Ribo Technologies) was applied to detect the colonization of bacteria in colonic mucosal tissues. In situ hybridization of bacteria was performed according to the manufacturer's instructions. Muc2 staining was performed followed by FISH staining. Briefly, the slides were blocked with 5% goat serum for 1 h at 37°C under dark. After blocking, tissue sections were incubated with anti–Muc2 primary antibody (GTX100664; GeneTex) for 2 h 37°C. Slides were washed three times (5 min for each time), with 0.1% TBST. Tissue sections were then incubated with anti-rabbit DyLight 488 (35552; Thermo

Fisher Scientific) 1 h at 37°C. After washing three times in TBST, DAPI was added on the tissue sections to stain the nucleus.

### Microbial DNA isolation and quantification

Fecal samples were collected from mice with or without TA or GA treatment and stored at –80°C until processed. Frozen fecal samples were processed for DNA isolation using QIAamp Fast DNA Stool Mini Kit (Qiagen). Real-time PCR was applied to measure the relative abundance of specific bacteria. The method for specific bacteria primer design was referred as previously described (Li et al, 2022). The primer of specific bacteria was summarized in Table S7.

### Metagenomic sequencing and data analysis

Stool was collected from TA-treated or untreated mice and extracted using PowerFecal DNA isolation kit (MOBIO). The Iso-Seq library was prepared according to the Isoform Sequencing protocol (Iso-Seq) using the Clontech SMARTer PCR cDNA Synthesis Kit and the BluePippin Size Selection System protocol as described by Pacific Biosciences (PN 100-092-800-03). Sequencing was per-formed to obtain DNA (125 bp) paired end reads to a depth of 10G base pairs per sample using HiSeq Illuminex 2500. Readfq (V8, https://github.com/cjfields/readfq) was used for preprocessing raw data from the Illumina sequencing platform to obtain clean data for subsequent analysis differential expression analysis of two groups was performed using the DESeq R package (1.18.0). DESeq provide statistical routines for determining differential expression in digital gene expression data using a model based on the negative binomial distribution. The resulting $p$-values were adjusted using the Benjamini and Hochberg's approach for controlling the false discovery rate. Genes with an adjusted $p$-value < 0.05 found by DESeq were assigned as differentially expressed. LEfSe software is used for LEfSe analysis based on the the abundance at each taxonomy level.

### Metabolomic analysis and data processing

Stools (100 mg) were individually grounded with liquid nitrogen, and the homogenate was resuspended with prechilled 80% methanol by well vortex. The samples were incubated on ice for 5 min, and then were centrifuged at 15,000$g$, 4°C for 20 min. Su-pernatant was diluted to final concentration containing 53% methanol by LC-MS–grade water. The samples were subsequently transferred to a fresh Eppendorf tube and then were centrifuged at 15,000$g$, 4°C for 20 min. The supernatant was injected into the LC-MS/MS system analysis.

UHPLC-MS/MS analyses were performed using a Vanquish UHPLC system (Thermo Fisher Scientific) coupled with an Orbitrap Q ExactiveTMHF-X mass spectrometer (Thermo Fisher Scientific) in Novogene Co., Ltd. The samples were injected onto a Hypesil Gold column (100 × 2.1 mm, 1.9 μm) using a 17-min linear gradient at a flow rate of 0.2 ml/min. The eluents for the positive polarity mode were eluent A (0.1% FA in water) and eluent B (methanol). The eluents for the negative polarity mode were eluent A (5 mM ammonium ac-etate, pH 9.0) and eluent B (methanol). The solvent gradient was set

as follows: 2% B, 1.5 min; 2–100% B, 3 min; 100% B, 10 min; 100–2% B, 10.1 min; and 2% B, 12 min. The Q ExactiveTM HF-X mass spectrometer was operated in positive/negative polarity mode with spray voltage of 3.5 kV, capillary temperature of 320°C, sheath gas flow rate of 35 psi, aux gas flow rate of 10 Liters/min, S-lens RF level of 60, and aux gas heater temperature of 350°C.

The raw data files generated by UHPLC-MS/MS were processed using the Compound Discoverer 3.1 (CD3.1; Thermo Fisher Scientific) to perform peak alignment, peak picking, and quantitation for each metabolite. The main parameters were set as follows: retention time tolerance, 0.2 min; actual mass tolerance, 5 ppm; signal intensity tolerance, 30%; signal/noise ratio, 3; and minimum intensity. After that, peak intensities were normalized to the total spectral intensity. The normalized data were used to predict the molecular formula based on additive ions, molecular ion peaks, and fragmentions. And then peaks were matched with the mzCloud (https://www.mzcloud.org/), mzVault, and MassList database to obtain the accurate qualitative and relative quantitative results. Statistical analyses were performed using statistical software R (R version R-3.4.3), Python (Python 2.7.6 version), and CentOS (CentOS release 6.6). When data were not normally distributed, normal transformations were attempted using the area normalization method. PCA was performed at metaX (a flexible and comprehensive software for processing metabolomics data). We applied univariate analysis ($t$ test) to calculate the statistical significance ($p$-value). The metabolites with VIP > 1 and $p$-value < 0.05 and fold change ≥ 2 or ≤ 0.5 were considered to be differential metabolites. Volcano plots were used to filter metabolites of interest which based on $\log_2$(fold change) and $-\log_{10}$($p$-value) of metabolites by ggplot2 in R language.

### RNA sequencing and analysis

Total amounts and integrity of RNA were assessed using the RNA Nano 6000 Assay Kit of the Bioanalyzer 2100 system (Agilent Technologies). Total RNA was used as input material for library construction. The library fragments were purified with the AMPure XP system (Beckman Coulter). After the construction of the library, the library was initially quantified by the Qubit 2.0 Fluorometer, then diluted to 1.5 ng/µl, and the insert size of the library is detected by the Agilent 2100 bioanalyzer. When the inserted size meets the expectation, qRT-PCR is used to accurately quantify the effective concentration of the library (the effective concentration of the library is higher than that of 2 nM) to ensure the quality of the library. After the library is qualified, the different libraries are pooling according to the effective concentration and the target amount of data off the machine, then being sequenced by the Illumina NovaSeq 6000. The end reading of 150 bp pairing is generated. The basic principle of sequencing is to synthesize and sequence at the same time (sequencing by synthesis). Four fluorescent-labeled dNTPs, DNA polymerase, and splice primers were added to the sequenced flow cell and amplified. When the sequence cluster extends the complementary chain, each dNTP labeled by fluorescence can release the corresponding fluorescence. The sequencer captures the fluorescence signal and converts the optical signal into the sequencing peak by computer

software, so as to obtain the sequence information of the fragment to be tested.

Differential expression analysis of two conditions/groups (two biological replicates per condition) was performed using the DESeq2 R package (1.20.0). DESeq2 provides statistical routines for determining differential expression in digital gene expression data using a model based on the negative binomial distribution. The resulting $p$-values were adjusted using the Benjamini and Hochberg's approach for controlling the false discovery rate. padj ≤ 0.05 and |$\log_2$(foldchange)| ≥ 1 were set as the threshold for significantly differential expression. We use the clusterProfiler R package (3.8.1) to test the statistical enrichment of differential expression genes in KEGG pathways. The Reactome database brings together the various reactions and biological pathways of human model species. Reactome pathways with corrected $p$-value less than 0.05 were considered significantly enriched by differential expressed genes. Gene Set Enrichment Analysis (GSEA) is a computational approach to determine if a predefined Gene Set can show a significant consistent difference between two biological states. The genes were ranked according to the degree of differential expression in the two samples, and then the predefined Gene Set were tested to see if they were enriched at the top or bottom of the list. GSEA can include subtle expression changes. We use the local version of the GSEA tool http://www.broadinstitute.org/gsea/index.jsp.

### HPLC–mass spectrometry

For tissue samples with or without TA and GA treatment, 0.1 g of samples were taken, homogenized, and centrifuged. Supernatants were collected and 200 µl of 50% methanol was added. After centrifuge, supernatants were collected for HPLC assay. For feces with or without TA and GA treatment, 0.06 g samples were collected and homogenized in 200 µl 50% methanol. Supernatants were then harvested for HPLC assay. For cultured feces, the liquid was collected and centrifuged. The supernatant was used for HPLC assay. The prepared supernatants were quantified using a Quantum Access MAX triple-stage quadrupole mass spectrometer.

### DSS-induced colonic colitis and treatment

Colonic colitis was induced in 6-wk-old C57BL/6 mice with 2.5% DSS (160110; MP Biomedicals) in drinking water. Mice were orally treated with different doses of TA (BD01418099; Bidepharm), GA (G7384; Sigma-Aldrich), and PBS (G4202; Servicebio) daily using a feeding needle. Stools of mice were collected for further analysis. This experiment was approved by the Ethics Committee of the Third Military Medical University. All procedures adhered to the guidelines approved by the Animal Experimentation Ethics Committee of the Army Medical University.

### FMT experiment

To analyze the effect of feces with TA treatment, 6-wk-old germ-free (GF) mice were gavaged with feces (in sterile PBS) collected from TA-treated mice twice (in a week). After that mice were left in the GF package for another week. Feces were collected to examine bacterial colonization. Mice were then administrated with 2.5% DSS.

This study was approved by the Animal Experimentation Ethics Committee of Army Medical University.

### Clinical score and histological analysis

Body weight was recorded and change of body weight was indicated as loss of baseline body weight as a percentage. Colon length was measured when mice were euthanized. Colon tissues was fixed with 4% PFA and embedded in paraffin. Sections of colon tissue were stained with H&E according to standard protocol. Histological scoring was assessed in a blind way by a pathologist. The scoring standard was referred as previously described (Bauer et al, 2010).

### Transmission electron microscope

Colon tissues (about 1 mm$^3$) of TA-treated mice were collected and fixed in 2.0% glutaraldehyde in 0.1 mol/liters sodium cacodylate (Electron Microscopy Sciences). Reichert Ultracut E ultramicrotome was used to prepare the ultrathin sections, which was then examined using a Philips CM100 transmission electron microscope.

### Measurement of tannase activity

Feces collected from TA- or GA-treated or control mice were measured for the activity of tannase. Tannase activity assay kit was used (BC4075; Solarbio), according to the manufacturer's instruction.

### Statistical analysis

All statistical tests were performed using GraphPad Software (Version 9.1). Data are presented as mean ± SD. Comparison between two groups was done using $t$ test. Kaplan–Meier analysis was applied to analyze the survival of GF mice. $p$-values < 0.05 were considered statistically significant.

## Data Availability

Data are available on reasonable request. All data are available from the corresponding author on request.

## Supplementary Information

## Acknowledgments

We thank the staff from Animal center of Army Medical University for caring the mice. This project was supported by the National Key Research and Development Program of China (NO.2021YFF0702300 and NO.2017YFD0500503).

## Author Contributions

Q He: data curation, software, formal analysis, validation, methodology, and writing—original draft.
K Guo: data curation, formal analysis, and methodology.
L Wang: software, formal analysis, and methodology.
F Xie: methodology and project administration.
Q Zhao: software.
X Jiang: methodology.
Z He: methodology.
P Wang: methodology.
S Li: methodology.
Y Huang: methodology.
C Zhang: methodology.
R Huang: methodology.
Y Liu: methodology.
F Wang: methodology.
X Zhou: methodology.
R Niu: funding acquisition.
T Zuo: conceptualization.
Y Wang: conceptualization, resources, data curation, supervision, funding acquisition, project administration, and writing—review and editing.
C Li: conceptualization, resources, data curation, software, formal analysis, supervision, validation, investigation, methodology, project administration, and writing—review and editing.

### Conflict of Interest Statement

The authors declare that they have no conflict of interest.

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
