## [Reviewer comments · Life Science Alliance]

Life Science Alliance

Tannins amount determines whether tannase-containing bacteria are probiotic or pathogenic in IBD

Qiuyue He, Kenan Guo, Lulu Wang, Fei Xie, Qingyuan Zhao, Xianhong Jiang, Zhongming He, Peng Wang, Shiqiang Li, Yan Huang, Cong Zhang, Rongjuan Huang, Yang Liu, Fengchao Wang, Xiaoyang Zhou, Rong Niu, Tao Zuo, Yong Wang and Chuangen Li

DOI: <https://doi.org/10.26508/lsa.202201702>

Corresponding author(s): *Dr. Chuangen Li (Army Medical University); Yong Wang*

Review Timeline:

Submission Date:	2022-09-01
Editorial Decision:	2022-10-20
Revision Received:	2023-01-09
Editorial Decision:	2023-01-16
Revision Received:	2023-02-01
Accepted:	2023-02-01

Scientific Editor: Novella Guidi

Transaction Report:

October 20, 2022

Re: Life Science Alliance manuscript #LSA-2022-01702-T

Dr. Chuangen Li
Army Medical University
Animal Laboratory Science
NO.30 Gaotanyan central street
Chongqing 400438
China

Dear Dr. Li,

Thank you for submitting your manuscript entitled "Tannins amount determines whether tannase-containing bacteria are probiotic or pathogenic in inflammatory bowel disease" to Life Science Alliance. The manuscript was assessed by expert reviewers, whose comments are appended to this letter. We invite you to submit a revised manuscript addressing the Reviewer comments.

Thank you for this interesting contribution to Life Science Alliance. We are looking forward to receiving your revised manuscript.

Sincerely,

B. MANUSCRIPT ORGANIZATION AND FORMATTING:

Reviewer #1 (Comments to the Authors (Required)):

The manuscript by He Q. et al. demonstrated that dietary tannin (TA) exerted either protective or pathologic effect on DSS-induced IBD depending on the amount of intake, and that gallic acid has a potential for prevention and recovery of IBD. The study is interesting and confirmed with a number of experiments. However, some information is missing. The reviewer has some comments as follows.

- 1) TAs are a mixture of polyphenols with high molecular weights. In this study, the information of TA used in this study is totally short. Where is it obtained? And, please indicate chemical structural formula of TA used in this study. Also, how were mice orally treated with TA, GA, and PBS. Using feeding needle or something? The detail information is necessary.
- 2) Also, the information of GA and DSS is short. Where is it obtained? Overall, the information of materials is quite short.
- 3) Table S2: It seems Soyasaponin 1 has higher FC compared with GA. The reviewer cannot understand why only GA was chosen. Please explain.
- 4) TA and GA showed the opposite effect on DSS-induced IBD depending on the concentration. Therefore, comparison between GA50 and GA250 for the metabolites in vivo as well as gene expression (RNAseq) in vitro might be important for the understanding of underlying mechanisms.
- 5) Figure 5G: The authors showed longer-term administration (30 days) of 250 ug of GA caused colonic inflammation. How the effect on longer-term administration of 50 ug of GA? In addition, did GA250 exacerbated the inflammation recovery experiment (Figure 7)?
- 6) The authors used *gadph* and *actb* for internal control in qPCR. However, several reports have shown that expression of these transcripts are altered by inflammation (ex BMC Immunol. 2017; 18: 43). Therefore, authors should re-analyze the reliable reference genes for the normalization of gene expression.
- 7) In manuscript (Result section), authors mentioned GA50 increased S-phase cell population (Figure 6F), but figure showed GA50 increased G0/G1.
- 8) There are several misspellings for the TA and GA, (ex. Figure legend 1C, 1D S1A). Figure legend S5 is missing. Figure 4B lacks the GA250 data. Figure S7C lacks GA50 data.
- 9) The authors demonstrated TA did not induce microbial alternation. In previous study, however, it is demonstrated that tannin could lead to microbial alternation using DSS-induced IBD model (Sci Rep. 2021 Mar 31;11(1):7286). Why are the results different from the authors' study? It should be discussed in the Discussion. And, overall, the Discussion seems enumeration of results. It should be edited.
- 10) Title: Overall, the data about TA250 and GA250 seems just toxic rather than pathogenic. Please explain in Discussion why it is not toxic, but pathogenic.

Reviewer #2 (Comments to the Authors (Required)):

The study is novel and provides possible new insights in the dietary management of IBD. With minor modification in the manuscript it could be accepted for publication

1. The running title could to be changed as it does not reflect the main title.
2. Cite recent references in the Introduction preferably 2016 onwards.
3. Include the source of Gallic acid and Tannic acid used in the study in the Material and methods section.
4. The results and discussion section is well described.

Reviewer #1 (Comments to the Authors (Required)):

The manuscript by He Q. et al. demonstrated that dietary tannin (TA) exerted either protective or pathologic effect on DSS-induced IBD depending on the amount of intake, and that gallic acid has a potential for prevention and recovery of IBD. The study is interesting and confirmed with a number of experiments. However, some information is missing. The reviewer has some comments as follows.

1) TAs are a mixture of polyphenols with high molecular weights. In this study, the information of TA used in this study is totally short. Where is it obtained? And, please indicate chemical structural formula of TA used in this study. Also, how were mice orally treated with TA, GA, and PBS. Using feeding needle or something? The detail information is necessary.

Response: Thank you for your advice. We bought TA from company. The information has been added in the "Material and Methods" part.

2) Also, the information of GA and DSS is short. Where is it obtained? Overall, the information of materials is quite short.

Response: Thank you for your advice. We bought GA from company. The information has been added in the "Material and Methods" part. We also added the information of other materials involved.

3) Table S2: It seems Soyasaponin 1 has higher FC compared with GA. The reviewer cannot understand why only GA was chosen. Please explain.

Response: Thank you for your question. Actually, we tested four candidates (FC>3.5, VIP>1, p<0.05, including Stearoyl ethanolamide, LPE, GA, and 10-Nitrolinoleate) using MTT method, which were described in the main text. Only GA displayed dual effects in the presence of DSS that it showed toxic effect in high concentration while it had protective effect in the low concentration, which was similar with the observation from the TA treated mice. Similar effect was not seen in other tested metabolites (Stearoyl ethanolamide and 10-Nitrolinoleate displayed promoting cell death while LPE displayed promoting cell proliferation).

4) TA and GA showed the opposite effect on DSS-induced IBD depending on the concentration. Therefore, comparison between GA50 and GA250 for the metabolites in vivo as well as gene expression (RNAseq) in vitro might be important for the understanding of underlying mechanisms.

Response: Thank you for your suggestions which are valuable. We previously performed RNA-seq using GA250 plus DSS. Because GA250 alone displayed toxic effect, we did not show the result of GA250+DSS. Here, we added this result in the RNA-seq part of the main text and the figure was added as Figure S9A and Table S6. We also changed the corresponding figure and table legends.

For the metabolites, we performed metabolomics following your advice and put this part of result in the main text. The results were uploaded in Figure S7 C & D and Table S3.

5) Figure 5G: The authors showed longer-term administration (30 days) of 250 ug of GA caused colonic inflammation. How the effect on longer-term administration of 50 ug of GA? In addition, did GA250 exacerbated the inflammation recovery experiment (Figure 7)?

Response: Thank you for your questions. We had the long term result of GA50 (**Figure S10D**), which showed only Muc2 was upregulated among tested genes, including inflammatory cytokines, chemokines, TJs proliferative and mucin genes.

For the recovery experiments, we only tested GA50, the protective dose. Because it is meaningful to show that this dose of GA promoted the recovery of inflammation. Moreover, after a 7-day 2.5% DSS administration, the mice were too weak, we worried about the death of mice with the toxic dose of GA treatment.

6) The authors used gadph and actb for internal control in qPCR. However, several reports have shown that expression of these transcripts are altered by inflammation (ex BMC Immunol. 2017; 18: 43). Therefore, authors should re-analyze the reliable reference genes

Response: Thank you for your suggestion. We re-analyzed some of the results with α -tubulin (Figure 1E&F, Figure 3E&F and Figure 7E), which showed similar trend with GAPDH or β -actin (see Figure 1 and 2 for reviewer). Besides, some studies published in

high-impact journals also used GAPDH or β -actin as the internal control, such as :

1 Luo, Y., Xie, C., Brocker, C. N., Fan, J., Wu, X., Feng, L., . . . Gonzalez, F. J. (2019). Intestinal PPARalpha Protects Against Colon Carcinogenesis via Regulation of Methyltransferases DNMT1 and PRMT6. *Gastroenterology*, 157(3), 744-759 e744

2 Sharma, D., Malik, A., Guy, C. S., Karki, R., Vogel, P., & Kanneganti, T. D. (2018). Pysin Inflammasome Regulates Tight Junction Integrity to Restrict Colitis and Tumorigenesis. *Gastroenterology*, 154(4), 948-964 e948.

3 Hernandez-Chirlaque, C., Aranda, C. J., Ocon, B., Capitan-Canadas, F., Ortega-Gonzalez, M., Carrero, J. J., . . . Martinez-Augustin, O. (2016). Germ-free and Antibiotic-treated Mice are Highly Susceptible to Epithelial Injury in DSS Colitis. *J Crohns Colitis*, 1

Since the trend was similar, we kept our original result with GAPDH or actin as the internal control. We put the partially re-analyzed WB and qPCR results in the figures for reviewer for your reference.

7) In manuscript (Result section), authors mentioned GA50 increased S-phase cell population (Figure 6F), but figure showed GA50 increased G0/G1.

Response: Thank you for finding this. We are sorry that the label was wrong, the red is GA50 treatment. We have revised it.

8) There are several misspellings for the TA and GA, (ex. Figure legend 1C, 1D S1A). Figure legend S5 is missing. Figure 4B lacks the GA250 data. Figure S7C lacks GA50 data.

Response: Thank you for your careful examination. We have double checked those mistakes and changed them. Sorry for the mistakes and inconvenience. Figure legend S5 is in supplementary methods and legends. For Figure 4B, because no difference was seen, we did not put it here at first and just mentioned the observation in the main text. On a second thought, we think it will be better if we put it here (changed in Figure 4B). Thank you. For Figure S7C, since GA (50 ug/ml) had strongly toxic effect alone, we did not performed the pre and non-pre treatment.

9) The authors demonstrated TA did not induce microbial alternation. In previous study, however, it is demonstrated that tannin could lead to microbial alternation using DSS-induced IBD model (Sci Rep. 2021 Mar 31;11(1):7286). Why are the results different from the authors' study? It should be discussed in the Discussion. And, overall, the Discussion seems enumeration of results. It should be edited.

Response: Thank you for your valuable suggestions. We added the discussion of different observations in the discussion. We also re-organized the discussion part.

10) Title: Overall, the data about TA250 and GA250 seems just toxic rather than pathogenic. Please explain in Discussion why it is not toxic, but pathogenic.

Response: Thank you for your question. We think you are right that our description is incorrect. The GA is toxic while the bacteria that produce GA are pathogenic. We have changed our description in the main text.

Reviewer #2 (Comments to the Authors (Required)):

The study is novel and provides possible new insights in the dietary management of IBD. With minor modification in the manuscript it could be accepted for publication

1. The running title could to be changed as it does not reflect the main title.

Response: Thank you for your suggestion. We have changed it from "Gallic acid is a potential drug for IBD treatment" to "Gallic acid possesses dual effects in IBD".

2. Cite recent references in the Introduction preferably 2016 onwards.

Response: Thank you for your suggestion. We updated the ref. 1 and 3. For Ref. 2 and 4, we did not change them as they are the original publication that describe the corresponding notion.

3. Include the source of Gallic acid and Tannic anic used in the study in the Material and methods section.

Response: Thank you for your careful examination. We have added the source of GA and TA in the Material and methods section.

4. The results and discussion section is well described.

Response: Thank you.

Figure 1 for reviewer

Figure 1 F

Figure3F

Figure7E

Figure 2 for reviewer

Figure 1 E

Figure 3 E

January 16, 2023

RE: Life Science Alliance Manuscript #LSA-2022-01702-TR

Dr. Chuangen Li
Army Medical University
Animal Laboratory Science
NO.30 Gaotanyan central street
Chongqing 400438
China

Dear Dr. Li,

Thank you for submitting your revised manuscript entitled "Tannins amount determines whether tannase-containing bacteria are probiotic or pathogenic in IBD". We would be happy to publish your paper in Life Science Alliance pending final revisions necessary to meet our formatting guidelines.

- please address the final Reviewer 1's comments
- please add ORCID ID for both corresponding authors-you should have received instructions on how to do so
- please consult our manuscript preparation guidelines <https://www.life-science-alliance.org/manuscript-prep> and make sure your manuscript sections are in the correct order
- please add a category for your manuscript to our system
- please add your supplemental methods section to the main materials and methods section; there isn't a word limit on this section
- please add your supplementary figure legends to the main manuscript
- please add the Twitter handle of your host institute/organization as well as your own or/and one of the authors in our system
- please use the [10 author names, et al.] format in your references (i.e. limit the author names to the first 10)
- please add a callout for Figure S8F to your main manuscript text

Figure Check:

- please add scale bars to Figure 2D, Figure S5B, and Figure S10A

A. FINAL FILES:

-- Summary blurb (enter in submission system): A short text summarizing in a single sentence the study (max. 200 characters including spaces). This text is used in conjunction with the titles of papers, hence should be informative and complementary to the title. It should describe the context and significance of the findings for a general readership; it should be written in the

present tense and refer to the work in the third person. Author names should not be mentioned.

B. MANUSCRIPT ORGANIZATION AND FORMATTING:

Sincerely,

Reviewer #1 (Comments to the Authors (Required)):

Overall, the revised manuscript was well improved. I have two comments before publication following the previous comments.

- 1) It is not still mentioned about how mice were orally treated with TA, GA, and PBS. Using feeding needle or something? It should be clearly mentioned.
- 2) Reference 20 in Discussion: Masahiro is maybe first name. It should be Kitabatake.

1) It is not still mentioned about how mice were orally treated with TA, GA, and PBS.

Using feeding needle or something? It should be clearly mentioned.

Response : Thank you for your suggestion. We used feeding needle and changed it in the maintext in the "Material and methods" part.

2) Reference 20 in Discussion: Masahiro is maybe first name. It should be Kitabatake.

Response: Thank you for finding this mistake. We have changed it as suggested in the "Discussion" part.

February 1, 2023

RE: Life Science Alliance Manuscript #LSA-2022-01702-TRR

Dr. Chuangen Li
Army Medical University
Animal Laboratory Science
NO.30 Gaotanyan central street
Chongqing 400438
China

Dear Dr. Li,

Thank you for submitting your Research Article entitled "Tannins amount determines whether tannase-containing bacteria are probiotic or pathogenic in IBD". It is a pleasure to let you know that your manuscript is now accepted for publication in Life Science Alliance. Congratulations on this interesting work.

DISTRIBUTION OF MATERIALS:

Again, congratulations on a very nice paper. I hope you found the review process to be constructive and are pleased with how the manuscript was handled editorially. We look forward to future exciting submissions from your lab.

Sincerely,
